

# The critical elements of the health system that could make for resilience in the World Health Organization African Region: a scoping review

Dick Chamla[1,*], Chinwe Iwu-Jaja[1,2,*], Anelisa Jaca[2], Asiphe Mavi Ndlambe[2], Muyunda Buwa[2], Ngozi Idemili-Aronu[3], Joseph Okeibunor[1], Charles Shey Wiysonge[1,2] and Abdou Salam Gueye[1]

[1] World Health Organization Regional Office for Africa, Brazzaville, Republic of the Congo
[2] Cochrane South Africa, South African Medical Research Council, Cape Town, South Africa
[3] University of Nigeria, Nsukka, Nigeria
* These authors contributed equally to this work.

Corresponding author
Chinwe Iwu-Jaja,
chinwelolo@gmail.com

## ABSTRACT

**Background:** Unpredictable events, such as the outbreak of infectious diseases and humanitarian crises, are putting a strain on health care systems. As a result, African countries will need to prepare themselves with appropriate strategies to withstand such occurrences. Therefore, the purpose of this scoping review was to map available evidence about what type and what components of health systems are needed to help countries cope with health emergencies and to foster health system resilience in the WHO African Region.

**Methods:** A systematic search was performed independently in Scopus and PubMed electronic databases as well as grey literature. Studies were selected based on set eligibility criteria based on the Joanna Brigg's Institute (JBI) methodology for scoping reviews. The key findings were focused on health system resilience and were mapped based on the WHO's core health system components. Our data were tabulated, and a narrative synthesis was conducted.

**Results:** A total of 28 studies were included in this scoping review, mostly conducted in the WHO African Region and region of the Americas. Studies focused on a variety of strategies, such as the continuous delivery of essential services, the strengthening of the health workforce, including community health care workers, community engagement, the provision of protective mechanisms for the health workforce, and flexible leadership and governance measures.

**Conclusion:** Our findings suggest that strategies to improve health system resilience must include all areas of the healthcare delivery process, including primary care. A resilient health system should be ready for a crisis and have adaptable policies in place to offer adequate response at all levels, as well as post-recovery planning. Such health systems should also seek for continuous improvement. More research is needed to assess the efficacy of initiatives for improving health system resilience, particularly in vulnerable African health systems.

## BACKGROUND

Health systems in countries have essential roles to play. Such functions include the ability of governments to respond swiftly and efficiently to infectious disease outbreaks with actions such as disease surveillance and control. Likewise, governments have the duty to mitigate the health impacts of other public health emergencies, including natural and man-made disasters (*Meyer et al., 2020*). Health systems are increasingly fragile due to unforeseen events (*Witter et al., 2017*), and we must reflect on the health system's readiness for these events (*Meyer et al., 2020*). The emergence and rapid spread of COVID-19 have also put a strain on the health systems around the world with a corresponding impact on the global economy (*Tessema et al., 2021*) and other aspects of life, including mental health and other social determinants of health (*Thomas et al., 2020*; *Alami et al., 2021*).

The effect of unforeseen shocks on health systems and services, including the enormous impact on morbidity and mortality, have been documented extensively following the outbreaks of infectious diseases such as the 2009 pandemic of influenza A (H1N1), the 2013–2016 West Africa Ebola epidemic (*Karamagi et al., 2022*) the 2015–2016 Zika outbreaks in Latin America and South-East Asia. These outbreaks showed how countries with weak health systems struggled to withstand these health system shocks (*Hasan et al., 2021*). Therefore, these emerging infectious disease outbreaks have awakened the need for strong and resilient health systems (*Hasan et al., 2021*; *Karamagi et al., 2022*). According to *Thomas et al. (2020)*, health system resilience is defined as the "ability to prepare for, manage (absorb, adapt and transform) and learn from shocks". The authors described shock as a "sudden and extreme change which impacts a health system" (*Thomas et al., 2020*). At the same time, there may be many definitions of health system resilience (*Fridell et al., 2020*; *Meyer et al., 2020*; *Hasan et al., 2021*; *Augustynowicz, Opolski & Waszkiewicz, 2022*); there is a common theme that cuts across these various definitions, which is the capacity or ability to cope with a catastrophe, handle it and rebuild the system, where necessary (*Fridell et al., 2020*; *Augustynowicz, Opolski & Waszkiewicz, 2022*). It also involves the ability of the health systems to continue to provide essential health services without interruption regardless of external events (*Karamagi et al., 2022*). Health system resilience is, thus, very crucial for the attainment of set health goals, such as universal health coverage (*Karamagi et al., 2022*).

### Rationale for conducting the study

The COVID-19 pandemic exposed the vulnerabilities and weaknesses inherent in global health systems, especially in Africa. Throughout history, the continent has faced various health crises, such as Ebola, HIV/AIDS, and other contagious diseases, which have consistently put pressure on its healthcare systems. These challenges emphasise the urgent

need for robust and resilient health systems that are capable of withstanding future disruptions.

While unforeseen events as described above destabilise a health system, they also open up an opportunity for improvements (*Burke et al., 2021*). The current discussions are centred around the impact of the COVID-19 pandemic, lessons from the responses, and how to prepare for and cope in the future (*Qijia Chua et al., 2020*; *Thomas et al., 2020*; *Wang et al., 2020*; *Alami et al., 2021*; *Burke et al., 2021*; *Arsenault et al., 2022*; *Talisuna et al., 2022*). Countries, therefore, need to develop strategies that will enable them to cope similar events, that may pose more threats in the long run (*Tessema et al., 2021*), which justifies our review. We decided to choose the scoping review methodology for this exercise to allows for comprehensive examination of the extent, range, and nature of research activity in this area, enabling us to identify gaps in the existing literature and to highlight opportunities for future research.

A preliminary search for any reviews or potentially eligible studies has shown research that focused on defining the key concept and a checklist that will aid in health system resilience (*Wulff, Donato & Lurie, 2015*; *Meyer et al., 2020*; *Alami et al., 2021*; *Augustynowicz, Opolski & Waszkiewicz, 2022*). Another review focused on how the integrated health service delivery model can benefit health system resilience (*Hasan et al., 2021*). Since this study was based on evidence as of 12 June 2020, we envisage more evidence would have emerged since then. There is a need for a study that looks at these key elements beyond health care delivery to other health system components. Furthermore, another review explored the health system's functionality, gaps, and critical elements before, during, and after emerging infectious diseases in low and middle-income countries (*Nuzzo et al., 2019*). However, their study was conducted before the COVID-19 pandemic. Hence there is a need to generate additional evidence within the context of the COVID-19 pandemic.

With the concept of health system resilience becoming a vital area requiring urgent attention, we need to understand how our health systems can become more resilient to address current challenges and better prepare for those that will come in the future (*Meyer et al., 2020*; *Hasan et al., 2021*). This is important to note that from a system-thinking perspective, all the components of a health system and their interactions are necessary to strengthen the system. This scoping review, therefore, aims to map available evidence that answers the following question: What interventions within the components of the health system are available to promote resilience in the WHO African Region in the context of infectious diseases capable of posing a serious threat to public health? We chose the scoping review methodology because it allows for a comprehensive examination of the extent, range, and nature of research activity in this area, enabling us to identify gaps in the existing literature and highlight opportunities for future research, including a systematic review which would be more specific research question (*Arksey & O'Malley, 2005*). This scoping review therefore helped to determine the extent of work done on strategies to build health system resilience, particularly in the context of emerging infectious diseases, the characteristics of the studies, and any gaps in the literature. Our findings will provide African policymakers with an overview of existing research on this topic and will assist

them in quickly gaining an understanding of this topic based on the most recent available research. Furthermore, any identified gaps can inform policymakers about areas where additional research is required.

Our study is intended for a diverse audience, including policymakers and public health officials, with a special emphasis on African nations. Additionally, healthcare practitioners, researchers interested in exploring health system resilience, particularly in the context of emerging infectious diseases, as well as global health organizations, will find this study valuable.

## METHODS

The Joanna Brigg's Institute (JBI) methodology was adapted for this scoping review (*Peters et al., 2015*, *2020*) and this review was reported according to the Preferred Reporting Items for Systematic Reviews and Meta-analyses extension for scoping review (PRISMA-ScR) (*Tricco et al., 2018*) (Appendix 1).

### Inclusion and exclusion criteria

The Population, Concept, and Context (PCC) framework described by *Peters et al. (2015*, *2020)* was used to determine studies eligible for inclusion. All eligible studies were included without restricting to specific populations. The concept of interest was "health system resilience," We considered studies that described any strategies used or can be used to strengthen or shield the health system against unforeseen circumstances. Any definition that fits into the earlier described definitions was included—"ability to prepare for, manage (absorb, adapt and transform) and learn from shocks" (*Thomas et al., 2020*). Our context was from the perspective of the COVID-19 pandemic and other infectious diseases that posed or are posing a serious threat to public health, especially those with epidemic potential.

For each included study, we identified strategies for enhancing resilience discussed in the article and mapped them based on the WHO's core components of the health system, namely: service delivery, health workforce, health information systems, medical products/vaccines/technology, financing, and leadership/governance (*World Health Organization, 2010*). We chose this framework for its comprehensive and widely recognized approach to analysing health systems. To further enhance the comprehensiveness of our mapping exercise we also adapted the conceptual framework developed by *Palagyi et al. (2019)*. This conceptual framework was developed from the traditional health system blocks, comprised of six components, focusing on the detection, prevention, and response to emerging infectious diseases. The framework includes six constructs organized into two primary themes. The first theme encompasses four constructs related to the "material resources and structures, or the system 'hardware,' which involves surveillance, infrastructure, medical supplies, workforce, and communication mechanisms". The second theme focuses on the "human and institutional relationships, values, and norms, or the system 'software,' which includes governance and trust".

## Types of sources

Due to the nature of our research questions and the current state of the COVID-19 pandemic as at when this article when written, we did not have restrictions on study design. Therefore, we considered all observational studies, including cross-sectional (descriptive and analytical), qualitative studies such as key informant interviews, and focus group discussions. Text and opinion articles, commentaries, and case studies were also considered for inclusion in this scoping review. Studies published in English language were included with no date limitations.

Studies that were not related to health system resilience were excluded from this study.

## Search strategy

An initial search of PubMed was undertaken to identify articles on the topic. The text words in the titles and abstracts of relevant articles used to describe the articles were used to develop a comprehensive search strategy (Appendix 2). The search strategy, including all identified keywords, was adapted for the Scopus database.

### Study selection

Following the search, all identified studies were collated and uploaded into Mendeley referencing software, where duplicates were automatically removed. The study selection was in two phases, with each phase undertaken in duplicates by two reviewers (AJ and CJ). The first stage involved a selection based on titles and abstracts only, while in the second phase, full-text articles were used to assess eligibility in potentially eligible studies. The full texts of selected articles were assessed in detail against the inclusion criteria by two reviewers (AJ and CJ). Full-text articles that did not meet the inclusion criteria were excluded. All disagreements between the reviewers at each stage of the selection process were resolved through discussion until a consensus was reached. The search results and the study inclusion process are presented in a PRISMA-flow diagram (*Page et al., 2021*).

## Data extraction

Data was extracted from articles eligible for inclusion by four reviewers (AJ, AMN, CJ, and MM) using a pre-designed data extraction tool. The data extracted included specific details: The study ID (authors' names and year of publication), study design, the country where each study was conducted, and key findings. The key findings were focused on health system resilience and were mapped based on the WHO's core health system components. All disagreements between the reviewers were resolved through discussion or with an additional reviewer/s.

## Data analysis and presentation

Our data were presented in a tabular form, and a narrative synthesis was conducted. The narrative summary describes the tabulated and charted results related to the review questions.
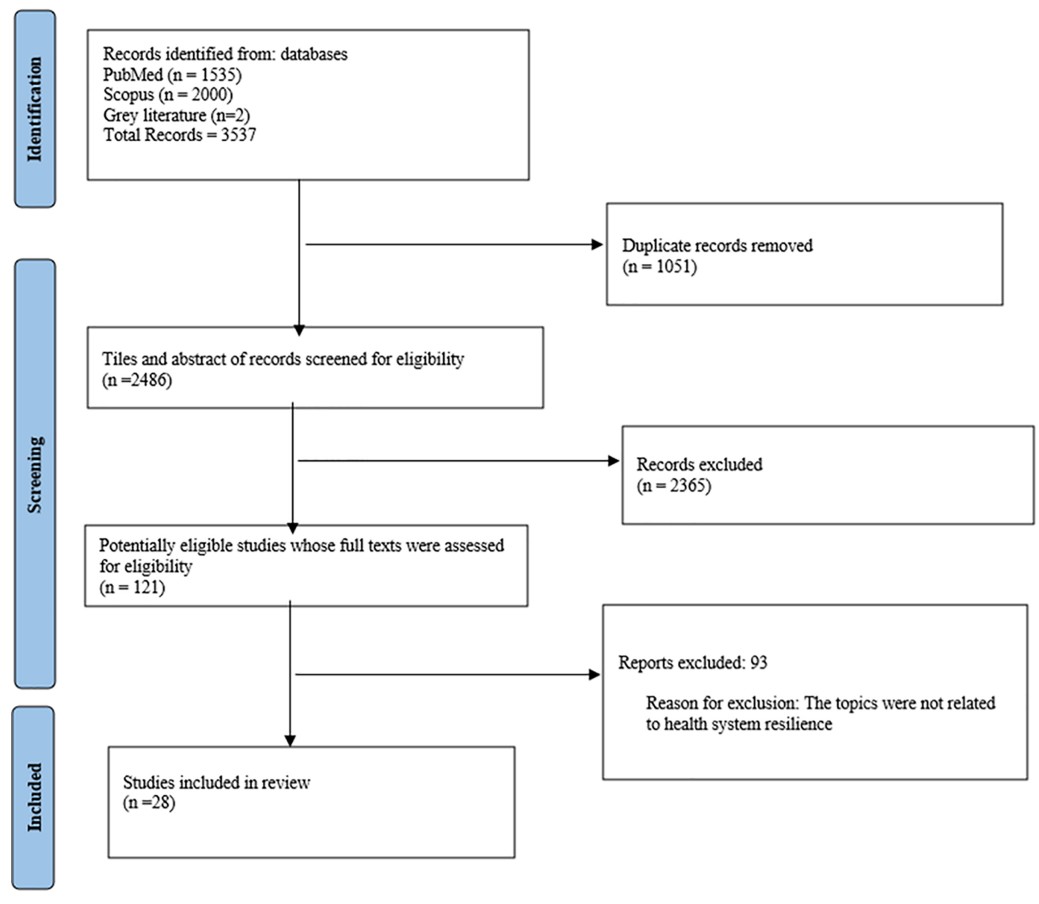

**Figure 1 PRISMA flow diagram showing the process of study selection.**

## RESULTS

### Results of the search

A total of 3,535 records were identified from searching PubMed and Scopus databases, and retrieved for screening. A total of 1,051 duplicates were removed, and the titles and abstracts of 2,484 articles were screened for potentially eligible studies. A total of 119 studies were deemed potentially eligible and their full texts were assessed for eligibility, and 28 articles were eventually included in the review. The study selection process is illustrated in Fig. 1.

### Characteristics of included studies

The 28 studies reporting strategies to achieve health systems resilience were conducted in four WHO regions. Seven studies (26.9%) were conducted in the WHO African Region (AFR) (*Witter et al., 2017*; *Adamu et al., 2020*; *Barker et al., 2020*; *Tumusiime et al., 2020*; *Marsh et al., 2021*; *Simen-Kapeu et al., 2021*; *Tidwell & Razak, 2021*). Seven studies (26.9%) were also conducted in the Region of the Americas (AMR); four in the United States of America (USA) (*Golden et al., 2021*; *Morganstein & Flynn, 2021*; *DeTore et al., 2022*; *Tebes et al., 2022*) and three in Canada (*Khan et al., 2018*; *Haldane et al., 2021*; *Leslie et al., 2021*;

*Pozzi et al., 2021*). Two studies (7.7%) were conducted in the South-East Asian Region (SEAR); one each in Bangladesh (*Nanda et al., 2020*) and India (*Garg et al., 2020*). In addition, two studies (7.7%) were conducted in the European Region (EUR), one involving multiple countries (Austria, Denmark, Germany, Italy, the Netherlands, and the United Kingdom (UK) (*Burau et al., 2022*) and the other in Ireland (*Burke et al., 2021*). Finally, one study (3.8%) was conducted in Hong Kong (China) in the Western Pacific Region (WPR) (*Lu, Li & Ni, 2021*). The remaining were either global in scope (*Lal et al., 2021*; *Wiig & O'Hara, 2021*; *Evans et al., 2022*; *Ghebreyesus et al., 2022*), focused on low and middle income countries (*Nuzzo et al., 2019*) or multi-country studies (*O'Sullivan et al., 2020*; *Hunte et al., 2020*; *Burau et al., 2022*).

This review included many study types/designs. These include a scoping review (*Nuzzo et al., 2019*) and other types of reviews (*Khan et al., 2018*; *Adamu et al., 2020*; *O'Sullivan et al., 2020*; *Hunte et al., 2020*; *Nanda et al., 2020*; *Garg et al., 2020*; *Pozzi et al., 2021*; *Tidwell & Razak, 2021*; *Wiig & O'Hara, 2021*; *Haldane et al., 2021*; *Leslie et al., 2021*; *Lu, Li & Ni, 2021*; *Morganstein & Flynn, 2021*; *Evans et al., 2022*; *Ghebreyesus et al., 2022*). The latter includes traditional reviews, commentaries, and editorials mostly emanating from lessons drawn from countries' experiences of various health system shocks and recommendations provided by authors. Eight studies included in this review were of qualitative design (*Barker et al., 2020*; *Tumusiime et al., 2020*; *Golden et al., 2021*; *Marsh et al., 2021*; *Burke et al., 2021*; *Burau et al., 2022*; *DeTore et al., 2022*; *Tebes et al., 2022*), while three employed mixed-methods study design (*Witter et al., 2017*; *Simen-Kapeu et al., 2021*; *Graetz et al., 2022*). Lastly, one study was a health policy brief (*Lal et al., 2021*).

We identified and classified the resilience strategies according to the six health systems' building blocks (Table 1). Some of these studies reported interventions targeted at multiple health systems' building blocks. We describe these classifications in more detail below.

## Summary of global evidence
### Service delivery

Nine studies reported interventions to strengthen resilience within service delivery (*Nuzzo et al., 2019*; *Adamu et al., 2020*; *Garg et al., 2020*; *Haldane et al., 2021*; *Leslie et al., 2021*; *Lu, Li & Ni, 2021*; *Pozzi et al., 2021*; *Graetz et al., 2022*). These interventions encompassed measures to guarantee uninterrupted provision of vital services, such as distributing infection prevention and control guidelines to primary care clinics to maintain their operations, constructing new treatment facilities, repurposing public venues, and reconfiguring existing medical facilities to accommodate COVID-19 patients, and utilising telemedicine for healthcare delivery. Additional measures involve establishing treatment facilities at the secondary or tertiary level of healthcare, equipped with designated isolation wards to effectively handle outbreaks of infectious diseases. A separate system to manage and treat infectious among confirmed cases should be established. Another intervention includes promoting community-based health services, including door-to-door counselling and community outreach to women to deliver services such as routine immunisation, contraceptive use, and infant feeding practices which can lead to improved health and nutrition behaviours.

**Table 1** Characteristics of included studies.

| Author ID | Country | Objectives | Study design | Type of health system shock/crises | Health system block | HSR strategy | Key findings/recommendations |
|---|---|---|---|---|---|---|---|
| *Adamu et al. (2020)* | Africa | A proposed framework for Africa's policymakers to implement the COVID-19 response within the context of the larger health system | Commentary | COVID-19 pandemic | The entire health system | **Response** | Response should be integrated focusing on the COVID-19 pandemic and other essential health services including childhood immunization, HIV treatment and other services during the pandemic. |
| *Barker et al. (2020)* | Liberia | To show the importance of community engagement (CE) for health system resilience | Qualitative study | Ebola epidemic | Communication | Response: community engagement Community engagement approaches: Information provision: passive transfer of information from health officials to communities; • Consultation: active exchange of information between health officials and communities, where community members' advice on planned interventions may be sought; • Participation: health officials and communities co-identify problems and implement solutions by empowering community structures or local institutions to deliver change; and community empowerment: health officials consult widely with and involve communities in local health-related decision-making and use community structures for service provision. | Community engagement led to improved communication with and increased trust in health authorities and programming. This facilitated health system response efforts, leading to a fortuitous cycle of increased trust, improved communication and continued meaningful CE. |
| *Marsh et al. (2021)* | Liberia | | **Qualitative study-Case study** | **COVID-19 pandemic** | Health service delivery  Workforce | Health workforce: telehealth and online learning including continued clinical learning for trainees  Community health workers stepped in to ensure continuity in the provision of essential services such as HIV, mental health disorders, non-communicable diseases. | Provision of HIV services continues without interruption; no reported COVID-19 cases among patients and the health workforce with significant improvement in programme performance. |

| Author ID | Country | Objectives | Study design | Type of health system shock/crises | Health system block | HSR strategy | Key findings/recommendations |
|---|---|---|---|---|---|---|---|
| | | | | | | | "The CHWs also helped clinics transition to multi-month dispensing of medications. CHWs led active community-based tracing for those lost to follow-up, and patients were successfully brought back to care after a high number of missed appointments in the early months of the pandemic". |
| *Witter et al. (2017)* | Uganda, Sierra Leone, Zimbabwe, and Cambodia | How health force cope with health system shocks | Mixed methods study-based on indepth interviews | Ebola epidemic (Sierra Leone); Economic crisis (Zimbabwe); and Political crisis; Conflicts targeted at health workers (Uganda and Cambodia) | Health workforce | Rebuilding health systems post-conflict: | Authors found that the impact of shocks and coping strategies are similar between conflict/post-conflict and epidemic contexts as it relates to physical and psychosocial threats |
| | | | | | | | Strategies adapted by this health workforce were categorized into physical survival strategies, psychosocial support and practical strategies relating to work and remuneration. |
| | | | | | | | Physical survival: self-protection, |
| | | | | | | | Psychosocial: family and community support |
| | | | | | | | Work and renumeration: additional earning options, dual practice, working longer hours, borrowing money, task shifting, using own resources for patients |
| | | | | | | | External support: provision of protective materials, training and workshops, donor support, additional allowance, rebuilding relationships with communities, |
| | | | | | | | Yet gaps still exists requiring that healthworforce should be better protected and supported |
| *Tumusiime et al. (2020)* | Africa | A report of a 2-day forum focused on building health system resilience to facilitate service continuity during health threats, primary health care revitalisation, and health systems strengthening towards UHC | Qualitative study-following stakeholder discussions | Multiple threats including COVID-19 pandemic | Service delivery | Building health system resilience | (1) Working cross-sectorally, (2) moving from fragmentation to integration, (3) ensuring implementation and knowledge exchange, and (4) rethinking resilience and embracing antifragility |

(Continued)

| Author ID | Country | Objectives | Study design | Type of health system shock/crises | Health system block | HSR strategy | Key findings/recommendations |
|---|---|---|---|---|---|---|---|
| *Tidwell & Razak (2021)* | Liberia | A report of a health initiative aimed at strengthening health workforce and the health system to be resilient and responsive to the Ebola epidemic including other health crises such as HIV/AIDs | Commentary | Ebola epidemic | Health workforce | **Creating a health workforce programme (HWP)** aimed at creating "a fit-for-purpose, motivated, and highly-skilled workforce." To cater for staff shortages. Through national and global support (partners)–The US Health Resources and Services Administration (HRSA) with the support from the US President's Emergency Plan for AIDS Relief (PEPFAR) | The programme after 5 years recorded the following achievements: Increasing the number of physicians, nurses, and midwives and their skills |
| **Simen-Kapeu et al. (2021)** | Liberia | This study reviewed the community health policy development process in order to draw lessons from the health system strengthening efforts in Liberia post-EVD crisis. | **Mixed methods study** | Ebola epidemic | Health workforce | Community health development programme: stablish a community health workforce that is incentivized to improve community-based service delivery in distant locations, and 2) ensure an enabling environment that restores faith in the health authority' ability to offer services through community involvement. | This research revealed the significance of key processes in the programme namely: Developing a coordination structure and leveraging partnership support, employing a systems approach to better inform policy adjustments, enhancing community engagement, and performing evidence-based planning to advise policymakers using a systems approach through a participatory process to better inform policy shifts; and strengthening community engagement and participation. Community engagement may have played a role in the decline in EVD transmission rates in Liberia |
| *Evans et al. (2022)* | Global | This study highlights the importance of life course immunisation | Commentary | COVID-19, current and future vaccine preventable diseases | Medical products-vaccination | Life course immunisation-process of vaccinating the entire population | Authors suggest that the approach used to vaccinate all individuals during the COVID-19 pandemic regardless of age could be adapted for other diseases. As this will help increase vaccination coverage and eventually contribute to health system resilience |
| *Graetz et al. (2022)* | USA, Philippines, and Spain | This study assessed how paediatric oncology teams and institutions in countries with differing resource levels navigated the COVID-19 pandemic | Mixed methods study | COVID-19 pandemic | **Service delivery** | Maintaining important services, like oncology treatment, | Strategies include reorganisation, the establishment of COVID-free zones, screening programmes, and interhospital collaboration to combat the spread of the virus |

## Table 1 (continued)

| Author ID | Country | Objectives | Study design | Type of health system shock/crises | Health system block | HSR strategy | Key findings/recommendations |
|---|---|---|---|---|---|---|---|
| *Burke et al.* (2021) | **Ireland** | The purpose of this study is to determine whether and how the Irish government's pandemic reaction contributed to the delivery of universal healthcare, health system reform, and greater resilience. | Qualitative study involving policy document analysis | COVID-19 pandemic | Governance | Pandemic response *via* health system reform programme known as Sláintecare to enhance universal healthcare. | Thirteen national policy documents revealed an increase in policy language and the intent to execute reform, as seen by an increase in policy alignment with and financial allocation to Sláintecare in conjunction with the implementation of major developments. |
| *Burau et al.* (2022) | Europe (Austria, Denmark, Germany, Italy, the Netherlands, and the UK) | To analyse the adaptable, absorptive, and transformative capacities of the health workforce during the first wave of the COVID-19 pandemic in Europe. | Qualitative study | COVID-19 pandemic | Health workforce | Ability to cope with the health crisis | To improve health workforce capacities and strengthen the integration of health professions in health governance. |
| *O'Sullivan et al.* (2020) | **Australia and Canada** | To present a perspective on the observations of the activity and experiences of the rural PHC sector during the COVID-19 epidemic that have been overlooked | Commentary | COVID-19 pandemic | Governance and policy | **Response to crisis** | Rural communities are different from their urban counterparts considering have dangers connected to movement and interaction patterns, population demands, socioeconomic disadvantage, and access and health service infrastructure. This needs localised risk assessment and communication. Pandemic resilience requires competent and stable PHC teams with flexible responses and resources to provide pandemic-related healthcare alongside primary care. This requires problem-solving with limited resources and employing networks and cooperation to serve huge geographic areas. PHC teams must secure methods for patient retrieval, equipment management, and staff relaxation. Rural PHC teams employ novel preventative clinics, screening, and ambulatory models to protect health workers while maximising population screening and continuity of care for vulnerable groups. Innovative pandemic models, such as telemedicine clinics, may inform rural health system capacity and patient access. |

(Continued)

| Author ID | Country | Objectives | Study design | Type of health system shock/crises | Health system block | HSR strategy | Key findings/recommendations |
|---|---|---|---|---|---|---|---|
| *Nuzzo et al.* (2019) | Low and middle income countries | A scoping review of the literature to identify recurring themes and capacities needed for health system resiliency to infectious disease outbreaks and natural hazards and any existing implementation frameworks that highlight these capacities. | Scoping review | Infectious diseases and natural hazards | All | Capacities needed for health system resiliency | 16 themes were identified as the capacities required: |

1) A resilient health system maintains routine healthcare delivery during a public health emergency

2) A resilient health system removes barriers to healthcare access so the public can receive emergency care

3) A resilient health system plans for critical infrastructure and transportation interruptions

4) A resilient health system has timely, flexible access to emergency/crisis financing to better prepare for and respond to public health emergencies

5) A resilient health system has a clear, flexible command structure that is exercised frequently

6) A resilient health system collaborates and coordinates with internal and external partners

7) A resilient health system has clear communication channels, risk communication protocols, and robust patient engagement.

8) A resilient health system has flexible plans and management structures to adapt to changing conditions

9) A resilient health system makes legal preparations for crises

10) A resilient health system can "surge" the level of care during public health emergencies

11) A resilient health system has adaptable response plans to allocate scarce resources and healthcare services

12) A resilient health system has a trained, willing workforce

13) A resilient health system has personal protective equipment, antivirals, and ventilators during a crisis

| Author ID | Country | Objectives | Study design | Type of health system shock/crises | Health system block | HSR strategy | Key findings/recommendations |
|---|---|---|---|---|---|---|---|
| | | | | | | | 14) A resilient health system has strong IPC measures, including staff training, standardised protocols, a dedicated IPC focal point, and dedicated treatment units |
| | | | | | | | 15) A resilient health system must commit to continuous quality improvement to promote excellence and earn community trust |
| | | | | | | | 16) Resilient health systems have plans for post-event recovery that address many issues |
| | | | | | | | However, no frameworks were found that translated these high-level themes into specific and actionable steps to improve and support health system resilience to infectious diseases and natural hazards in low-, middle-, and high-income settings. |
| DeTore et al. (2022) | USA | To examine the feasibility and acceptability of a brief online course focused on introducing evidence-based skills that could increase resilience and decreases emotional distress in healthcare workers during the pandemic. | Qualitative | Covid-19 pandemic | Health workforce | **Resilience Training:** (RT) course (pre-recorded videos): 1. focused on the concept of resilience and mindfulness skills | One and two months later, substantial increases in resilience and decreases in emotional distress were found among individuals who completed all three evaluations and took the course ($n = 38$), as compared to those who did not. |
| Golden et al. (2021) | USA | The Center for Stress, Resilience, and Personal Growth (CSRPG) was created to address the current and anticipated psychological impact of the pandemic on the HCWs in the health system. The mission of the Center is to support the resilience and mental health of employees through educational offerings, outreach, and clinical care. Our aim was to build a mobile app to support the newly founded Center in its mission. | Qualitative | Covid-19 pandemic | Health workforce | **Training:** through a digital platform (appl) that hosts a suite of tools that users can interact with daily. | The Wellness Hub app is a promising proof of concept, with room to grow, for those who wish to build a secure mobile health app to support their employees, communities, or others in managing and improving mental and physical well-being. It is a novel tool offering mental health support broadly. |

| Author ID | Country | Study design | Objectives | Type of health system shock/crises | Health system block | HSR strategy | Key findings/recommendations |
|---|---|---|---|---|---|---|---|
| *Tebes et al. (2022)* | USA | Qualitative | To describe an intervention to support the health workforce and summarize results from its 40-week implementation in a large, tri-state health system during the COVID-19 pandemic. | Covid-19 pandemic | Health workforce | **Support:** getting support from friends and family. | The virtual and interactive Stress and Resilience Town Hall is an accessible, scalable, and sustainable intervention to build mutual support, wellness, and resilience among healthcare workers and within hospitals and health systems responding to emerging crises, pandemics, and disasters. |
| *Leslie et al. (2021)* | Canada (Alberta) | Review | This article begins by distinguishing the technical and political aspects of resilience and then draws on a documentary analysis and qualitative interviews with health system and PC stakeholders to examine competing resilience-focused responses to the pandemic in Alberta, Canada. | Covid-19 pandemic | Service delivery | **Communication:** disseminate infection prevention and control guidance to PC clinics and the improvisational efforts of staff at those independent clinics to operationalize the guidance and ensure continuity of operations. | As this policy planning and structure design occurs, the goal is to leverage existing linkages between central service delivery systems and independent PC in ways that avoid the inconsistencies and potential burnout of responses that rely too heavily on individuals and improvisation. |
| *Lal et al. (2021)* | Global | Health policy article | This Health Policy article compares three types of health systems (*i.e.*, with stronger investments in global health security, stronger investments in universal health coverage, and integrated investments in global health security and universal health coverage) in their response to the ongoing COVID-19 pandemic and synthesises four essential recommendations (*i.e.*, integration, financing, resilience, and equity) to reimagine governance, policies, and investments for better health towards a more sustainable future. | Covid-19 pandemic | Leadership/governance<br><br>Financing | **Review preparedness models:** such as the WHO Joint External Evaluations and Service Availability and Readiness Assessments, can be reviewed and pursued together in resilience models, alongside consideration of social determinants of health to assess effects on health inequities, to develop a cohesive understanding of GHS and UHC gaps in health governance.<br><br>**Health financing:** A reimagined framework for global health that prioritises health-system integration across UHC and GHS domains, innovative and unified health financing, cross-sector resilience indicators, and equity as a core value offers a necessary path ahead. | National authorities developing health-system priorities and funders, who control expenditure, agenda setting, and prioritisation of investment, cannot continue business as usual. To rebuild a more sustainable future after COVID-19, embedding the core capacities of GHS into holistic, publicly financed UHC systems is the clear next step forward. |

| Author ID | Country | Objectives | Study design | Type of health system shock/crises | Health system block | HSR strategy | Key findings/recommendations |
|---|---|---|---|---|---|---|---|
| *Wiig & O'Hara (2021)* | Global | The commentary aims to outline main challenges and opportunities in resilient healthcare theory and practice globally, as a backdrop for contributions to the collection | Review | Covid-19 pandemic | Health information systems | **Digitization** of healthcare. **Communication:** communicating to diverse stakeholder groups. | Digitalization of healthcare, security issues, pandemics open possibilities for modelling resilience in new ways. To communicate the sometimes-complex message of resilience research to diverse stakeholder groups, as these are both target audiences and key actors for resilient healthcare system and services. Strategies and interventions to strengthen resilience in health systems and in service provision should be research-based. |
| *Lu, Li & Ni (2021)* | Hong Kong | Highlighting the challenges that Covid-19 brought and the interventions such as physical resilience, as well as psychological and mental resilience, by means of individual- or community-based health services or online psychological support could be employed as an adjuvant therapy for coping with the pandemic. | Review | Covid-19 pandemic | Service delivery | **Promoting community-based health services:** this should be a wake-up call for the government to pay special attention to the individuals with lower SES and old age, because these people may be the real vulnerable ones hidden behind the age-specific distribution. Therefore, promoting and strengthening physical resilience, as well as psychological and mental resilience, by means of individual- or online psychological support could be employed as an adjuvant therapy for coping with the pandemic. | Beside of routine treatment, developing an adaptive management for enhancing the resilient response to COVID-19 pandemic should be integrated into healthcare services, not just in the acute phase, but also in the post-COVID future. |
| *Morganstein & Flynn (2021)* | USA | To discuss current and emerging literature on the unique impacts of COVID-19 on HCWs and provides actionable, evidence-informed recommendations for individuals, teams, and leaders to enhance sustainment of HCWs that is critical to the preservation of national and global health security. | Review | Covid-19 pandemic | Leadership/governance Health workforce Medical products/vaccines/technology | **Modelling Self-Care:** leaders who model crisis behaviours give permission to their personnel to do the same. Taking steps to demonstrate desired behaviour to subordinates, encouraging supervisors at all levels to do the same, and instituting processes and procedures that facilitate self-care actions strengthen a workforce. **Practical Supports:** food, parking, lodging, and child-care are essential to allow HCWs to focus on work duties and reduce absenteeism and presenteeism. | This study concluded that it is important that interventions involve actions for individuals (and their families), organizations, and leaders to ensure an effective "whole of healthcare" approach. Health systems will benefit from borrowing and adapting existing practices from other high-risk occupations developed to improve functioning and optimize sustainment during prolonged crisis events. |

(Continued)

| Author ID | Country | Study design | Objectives | Type of health system shock/crises | Health system block | HSR strategy | Key findings/recommendations |
|---|---|---|---|---|---|---|---|
| | | | | | | **Family Safety:** procedures to reduce infection risk allow HCWs to feel comfortable coming to work and reduce stress for them and their family members. **Training:** timely, thorough, realistic, and updated training that prepares HCWs for anticipated exposures optimizes safe task performance and reduces stress of uncertainty.<br><br>**Equipment:** providing adequate supplies of effective equipment to protect HCWs for which they are adequately trained to use enhances perceived safety and reduces risk to HCWs and patients.<br><br>**Communication:** timely, regular, updated, truthful messages that articulate what is known, what is not known, and create realistic times for sharing additional information will enhance trust in organizations and leaders, improve compliance with recommendations, and optimize functioning. | |
| *Haldane et al. (2021)* | Canada | Review | To report on domains addressing governance and financing, health workforce, medical products and technologies, public health functions, health service delivery and community engagement to prevent and mitigate the spread of COVID-19. We then synthesize four salient elements that underlie highly effective national responses and offer recommendations. | Covid-19 pandemic | Health workforce<br><br>Medical products/vaccines/technology<br><br>Service delivery<br><br>Financing | **Health workforce reallocation and recruitment:** reviewed countries reallocated healthcare professionals, including primary-care workers, to emergency care wards, intensive care units (ICUs) and diagnosis and surveillance activities. Several recruitment strategies were implemented to increase the healthcare workforce. Retired, student or nonpracticing medical and paramedical professionals were asked to volunteer for healthcare tasks. For example, medical and nursing students were recruited and allowed to perform supervised work in different COVID-19 response capacities in countries such as Germany, Russia, Spain, the United Kingdom, and Vietnam. | Covid-19 responses in 28 countries: using our framework, we organized our results beginning with domains often viewed as external to health, which are nevertheless central determinants of health systems resilience—governance, finance, collaboration across sectors and community engagement—before exploring domains more closely associated with traditional views of health and health systems—health service delivery, health workforce, medical products and technologies and public health functions. |

| Author ID | Country | Objectives | Study design | Type of health system shock/crises | Health system block | HSR strategy | Key findings/recommendations |
|---|---|---|---|---|---|---|---|
| | | | | | | **Medical products and technologies:** high-quality prevention, diagnosis and management of COVID-19 require the ongoing development, production, and sustained distribution of mass quantities of medical products and technologies.<br><br>**Health service delivery:** health systems globally have employed three common approaches to rapidly scale up health system infrastructure, namely by constructing new treatment facilities, converting public venues and reconfiguring existing medical facilities to provide care for patients with COVID-19.<br><br>**Financing health systems:** while the relative importance of the various strategies and their configurations will depend on the specific country contexts, governance emerges as the foundation and lever for health system functioning and resilience. It plays a crucial role in enabling all other functions to work in unison to ensure adequately financed and otherwise well-resourced health service delivery to promote improved health.<br><br>**Public health functions:** public health interventions embedded within communities, such as testing, contact tracing, quarantine or self-isolation, and surveillance are crucial functions to break chains of transmission.<br><br>**Community engagement:** deep engagement with local communities is central to resilient health systems to inform service delivery, decision-making and governance and to meet the needs of communities before, during and after crises. Community engagement strategies, such as building partnerships with local leaders and working alongside community members to tailor messages and campaigns are crucial during public health emergencies. | |

(Continued)

| Author ID | Country | Study design | Objectives | Type of health system shock/crises | Health system block | HSR strategy | Key findings/recommendations |
|---|---|---|---|---|---|---|---|
| *Pozzi et al. (2021)* | USA | Review | To reflect on our standard delivery of care in thoracic surgery and to apply lessons learned during the pandemic to "rethink" how we optimize resources and deliver excellent patient care. | Covid-19 pandemic | Service delivery | **Telemedicine:** during this unprecedented time, new methods of healthcare delivery such as telemedicine have been developed, allowing us unique opportunities to reflect on our standard delivery of care in thoracic surgery and to apply lessons learned during the pandemic to "rethink" how we optimize resources and deliver excellent patient care. | The COVID-19 pandemic, despite is lethality, is no exception. Leadership that invokes innovation and nimbleness both in command structure and implementation is the hallmark of the many amazing stories, outcomes and heroic saves that COVID-19 delivered. These qualities are part of a highly reliable organization. Yet these qualities only lead to outstanding performance and executive function when this culture is coupled with highly dedicated and highly trained people. It is the people that make it happen. People who work together. |
| *Garg et al. (2020)* | India | Review | Highlighting lessons needed to be learnt especially for developing economies like India where public healthcare system is grossly inadequate to take care of health needs of citizens. | Covid-19 pandemic | Leadership/governance  Health information systems  Health workforce  Financing  Medical products/vaccines/technology  Service delivery | **Enforcement of many acts and laws:** Covid-19 pandemic in India led to address governance issues and democratic and fundamental rights of citizens were restricted with the promulgation of these acts, for example, Epidemic Diseases Act, Disaster Management Act, Essential Commodities Act, Healthcare Establishment Act, *etc*, It needs to be understood that many of such acts have their own inherent flaws and need to be modified for the current context.  **Telemedicine:** m-health & digital platforms.  **Basic knowledge of public healthcare system:** of the country, epidemiology, programs available for improving health of community, *etc*, are given to an undergraduate. This is essential to make him a competent primary care physician. Postgraduation is offered as Master's Degree (MD) in Community Medicine with training in epidemiology, disease surveillance, health systems, health programs, and public health laws. Master students are trained additionally on research on public health problems of the country. | Well-equipped and staffed Health and Wellness centers can contribute to close surveillance of outbreak in catchment area and aid response measures. |

| Author ID | Country | Objectives | Study design | Type of health system shock/crises | Health system block | HSR strategy | Key findings/recommendations |
|-----------|---------|------------|--------------|-----------------------------------|---------------------|--------------|------------------------------|
| | | | | | | **National Health Protection Scheme:** is envisaged to provide financial risk protection to poor and vulnerable families arising out of secondary and tertiary care hospitalization. | |
| | | | | | | **An overall Hospital Emergency Response Plan:** including an Epidemic Sub-plan, along with an Incident Command Group to coordinate the hospital's overall emergency response, should be available. SOPs should be in place detailing on the supply chain for acquiring, stocking, and, distributing the necessary supplies in the quantities required before and during an emergency and ensure that these procedures are consistent with national policies and national emergency response plans. | |
| | | | | | | **Treatment facility** | |
| | | | | | | **The Ayushman Bharat scheme:** should incorporate elements to address surge capacity at the time of health emergencies and measures to deliver care at the time of digital platforms or apps should contribute to trainings, supervision, and facilitation of healthcare delivery at remote locations. | |
| | | | | | | **Open data sharing policies:** should be developed for the practice of evidence-based public health. | |
| | | | | | | **Training:** give a boost to public healthcare system and health manpower trained in epidemiology to have system readiness to respond in case of future pandemics. | |

| Author ID | Country | Study design | Objectives | Type of health system shock/crises | Health system block | HSR strategy | Key findings/recommendations |
|---|---|---|---|---|---|---|---|
| *Hunte et al.* (2020) | Trinidad & Tobago, Spain | Review | To assess countries based on four of the six WHO criteria for rolling back "lockdown" measures: transmission controlled, test/trace/isolate, manage risk of exporting and importing cases, and community fully engaged. | Covid-19 pandemic | Leadership/governance<br><br>Health workforce<br><br>Financing<br><br>Medical products/vaccines/technology<br><br>Service delivery | **Key mitigation and containment strategies:** reflect the "all-of-government" approach. These include data-driven, evidenced-informed decision-making and guidelines from the Ministry of Health; the provision of financial resources from the Ministry of Finance; border/immigration restrictions implemented by the Ministry of National Security; remote work and pandemic leave policies prepared by the Ministry of Labor and Small Enterprise Development; the provision of social support services by the Ministry of Social Development and Family Services; and communications coordinated by the Ministry of Communications.<br><br>**Add staff:** Specific to the pandemic, the projected need for additional staff was assessed, and measures were taken to address them, including invitation of retired health professionals to express their interest and availability to be contracted should there be a surge of COVID-19 cases, and addressing the shortage of intensive care unit (ICU) nurses by a special arrangement with the University of the West Indies for the training of local nurses and recruitment of ICU nurses from Cuba. The protection of frontline personnel was paramount, and they were provided with adequate and appropriate PPE, as well as quarantine facilities before returning to their families after working with COVID-19 patients. The Ministry also ensured psychological support was available for staff. | Ensuring that key mitigation and containment strategies were well-coordinated, collaborative, evidence-informed, and timely was critical to the success of the T&T health system response to COVID-19. This was especially so for the leadership and governance and service delivery functions. The country's health system response is a reminder that even in developing countries, fraught with many health systems challenges, a combination of political will, decisiveness, respect for science, and the utilization of evidence-informed policies can have positive outcomes for populations during a health crisis. As the country rolls out its phased reopening, including the reopening of its borders, the expectation is that the underpinning principles and actions that contributed to the initial successful containment of COVID-19 will be sustained. |

| Author ID | Country | Objectives | Study design | Type of health system shock/crises | Health system block | HSR strategy | Key findings/recommendations |
|-----------|---------|------------|--------------|-----------------------------------|---------------------|--------------|------------------------------|
| | | | | | | **Increase budget:** the Ministry of Finance identified, and the Cabinet approved increased budgetary allocations to various levels of the health sector, the Ministry of Health, Regional Health Authorities (RHAs), and the main procurement agency, the National Insurance Property Development Company Limited. There was also an assurance that additional requests for funding related to the pandemic would be considered and prioritized by the Ministry of Finance.<br><br>**Procuring supplies:** The Ministry and RHAs were also supported in procuring supplies through its regional and international partners, *i.e.* the Pan American Health Organization and the United Nations Development Programme. In addition, several local manufacturers and other agencies bolstered availability with the local production of supplies for healthcare workers, including hand sanitizers, face shields, and face masks.<br><br>**Containment strategies:** implemented and communicated to the public was the establishment of a parallel health system, that is, a separate system to manage and treat confirmed COVID-19 cases, independent of the routine public health system.<br><br>**Funding:** there was an assurance that additional requests for funding related to the pandemic would be considered and prioritized by the Ministry of Finance.<br><br>**Communication:** dedicated COVID-19 hotlines were established to address queries from the public and provide initial screening and early guidance on safely accessing health care. | |

| Author ID | Country | Objectives | Study design | Type of health system shock/crises | Health system block | HSR strategy | Key findings/recommendations |
|---|---|---|---|---|---|---|---|
| *Nanda et al. (2020)* | India | To reflect on the opportunities and implications for leveraging this momentum to build a more resilient, gender equitable and community-engaged public health system. | Review | Covid-19 pandemic | Service delivery | **Community outreach:** evidence shows that door-to-door counselling and community outreach to women does lead to improved health and nutrition behaviours and service uptake such as routine immunisation, contraceptive use, and infant feeding practices. **Communication:** CHWs have been galvanised to play a range of roles: raising awareness and disseminating information on preventive measures; engaging in community surveillance, including contact tracing; conducting door-to-door surveys to assess returning migrants; delivering take-home rations; and addressing myths and misconceptions to mitigate stigma and discrimination | Effort needs to be drawn on available evidence and insights to sustain and build on the progress seen in the last few months. Notably, lauding CHWs for their efforts is a step in the right direction. However, along with greater recognition, a rationalisation of their workload is in order. The COVID-19 benefit package is a positive step but is time-bound; streamlining CHWs' remuneration to ensure fair compensation and greater economic security1 can help boost their status within the health system, community, and families. Adoption of supportive supervision can help CHWs overcome performance challenges (including those caused by system constraints) and empower them to perform. 38 Training that is fit for purpose, prompt, and covers transferable skills such as communication is essential to improve both performance and motivation. Lastly, CHWs should include male and female health workers accorded with equal professional skills and status. Feminising the workforce of community health workers and keeping them underpaid or unappreciated does not bode well for gender equality or enabling universal health coverage |
| *Khan et al. (2018)* | Canada | To describe the critical components of a resilient public health system and how they interact as a complex adaptive system. | Qualitative study using focus group discussion | All types of emergencies and disasters ("all hazard emergencies" including infectious diseases, flooding, wild fires, explosions | Governance and leadership | General public health emergency preparedness | The authors proposed a framework for public health emergency preparedness, with ethics and values at its core, and emphasised the system's complexity. |

| Author ID | Country | Objectives | Study design | Type of health system shock/crises | Health system block | HSR strategy | Key findings/recommendations |
|---|---|---|---|---|---|---|---|
| Ghebreyesus et al. (2022) | Global | A position article aimed at highlighting strategies that countries should adapt to build resilient health systems | Commentary (position article) | COVID-19 | Multidisciplinary with no specific focus on health system building blocks | Intersectorial actions covering health, finance and other sectors | "(i) leverage the current response to strengthen both pandemic preparedness and health sys-tems; (ii) invest in essential public health functions including those needed for all-hazards emergency risk management; (iii) build a strong primary health-care foundation; (iv) invest in institutionalized mechanisms for whole-of-society engagement; (v) create and promote enabling environments for research, innovation and learning; (vi) increase domestic and global investment in health system foundations and all-hazards emergency risk management; and (vii) address pre-existing inequities and the disproportionate impact of CO-VID-19 on marginalized and vulnerable populations" |

### Health workforce

A total of eight studies reported strategies ranging from training healthcare workers to increase their capacity for response, to connecting with colleagues, or supervisors, getting support from family or friends, and increasing the healthcare workforce (including recruitment of healthcare professionals such as primary-care workers), to emergency care wards and intensive care units (ICUs) and diagnosis and surveillance activities, increasing the healthcare workforce (*Hunte et al., 2020*; *Garg et al., 2020*; *Golden et al., 2021*; *Haldane et al., 2021*; *Morganstein & Flynn, 2021*; *DeTore et al., 2022*; *Tebes et al., 2022*). Another strategy involves boosting the role of health professionals in health governance (*Burau et al., 2022*).

### Health information systems

Of the 26 included studies, four reported on digitalizing healthcare to improve health systems resilience (*Garg et al., 2020*; *Wiig & O'Hara, 2021*), including telemedicine (*O'Sullivan et al., 2020*; *Marsh et al., 2021*).

### Access to medical products, diagnostics, and essential medicines

Four studies reported interventions involving providing enough and effective equipment to protect HCWs, adequate medical products such as hand sanitizers, face shields, and face masks and technologies (*Hunte et al., 2020*; *Garg et al., 2020*; *Haldane et al., 2021*; *Morganstein & Flynn, 2021*). Another strategy suggests a life course vaccination approach as a potential contributor to health system resilience (*Evans et al., 2022*), using the strategy employed to vaccinate all persons regardless of age during the COVID-19 epidemic.

### Financing

Only two of the included studies reported on funding, increasing budgetary allocations to health sectors, and improving financing structure (*Nuzzo et al., 2019*; *Hunte et al., 2020*; *Garg et al., 2020*). In addition, financing essential health services required for preparedness, investing in primary care were reported in a WHO position article (*Ghebreyesus et al., 2022*).

### Leadership or governance

Strategies reported include modification of acts and laws that address governance issues and democratic fundamental rights; using evidenced-informed decision-making and guidelines; implementing remote work and pandemic leave policies prepared for self-care and strengthening of the workforce; providing social support services, and communicating with local communities to address service delivery, decision-making and governance and to meet the needs of communities before, and during health crises. In addition, communication involves working with community members to pass messages and campaigns during public health emergencies, addressing inequities (*Khan et al., 2018*; *Nuzzo et al., 2019*; *Hunte et al., 2020*; *Garg et al., 2020*; *Lal et al., 2021*; *Morganstein & Flynn, 2021*; *Burke et al., 2021*).

Other strategies include communication and patient engagement, collaborations with both internal and external bodies, having adaptable plans and management structures in place, being committed to continuous improvement, and having post-recovery plans in place (*Nuzzo et al., 2019*).

## Summary of evidence from Africa

We found seven studies (*Witter et al., 2017*; *Adamu et al., 2020*; *Barker et al., 2020*; *Tumusiime et al., 2020*; *Marsh et al., 2021*; *Simen-Kapeu et al., 2021*; *Tidwell & Razak, 2021*), most of which focused on health system resilience in the context of infectious diseases, mostly COVID-19, the Ebola Epidemic, and HIV. Three health system strategies reported were mostly focused on service delivery (*Adamu et al., 2020*; *Tumusiime et al., 2020*; *Marsh et al., 2021*), while the remaining focused on the health workforce (*Witter et al., 2017*; *Marsh et al., 2021*; *Simen-Kapeu et al., 2021*; *Tidwell & Razak, 2021*) and governance (*Adamu et al., 2020*) and communication (*Barker et al., 2020*). Two studies were reviews (*Adamu et al., 2020*; *Tidwell & Razak, 2021*), three were qualitative studies (*Barker et al., 2020*; *Tumusiime et al., 2020*; *Marsh et al., 2021*) and two, mixed methods studies (*Witter et al., 2017*; *Simen-Kapeu et al., 2021*).

### Service delivery

Four studies emphasized the need for continuous provision of essential services during a health crisis, with the integration of a community-based approach. One of these studies proposed a framework that ensures a more comprehensive health systems-based model for COVID-19 outbreak response rather than an isolated response (*Adamu et al., 2020*). Clear and focused policymaking, sufficient operational ability to implement these policies utilising existing inputs and processes, and gaining public trust and stakeholder support through a community-driven approach were the three main tenets of this framework (*Adamu et al., 2020*). In addition, *Marsh et al. (2021)* presented a case study of how Liberia continued to make progress toward their HIV goals (95-95-95) while responding to the COVID-19 pandemic. They adopted several approaches, including training their health workforce, ensuring adherence to safety protocols, providing relevant PPEs, and using telehealth (*e.g.*, using telephones to follow up on patients) (*Marsh et al., 2021*). In the context of the Ebola epidemic in Liberia, a qualitative study demonstrated the significance of employing a community-based strategy to address health concerns. This study showed that community involvement in information dissemination, problem identification, intervention formulation, and health-related decision-making enhanced trust in health authorities and programmes (*Barker et al., 2020*).

### Governance, communication

Similarly, lessons from a community health development programme were presented in a mixed methods study in Liberia. This program aimed to create and incentivise the community health workforce to enhance community-based service delivery, including community engagement in remote locations to foster community trust (*Simen-Kapeu et al., 2021*).

### Other areas that emanated from our analysis

Furthermore, a review emphasised the need for response strategies to also cater to primary health care, often neglected. Authors stressed that challenges faced within this context are different, requiring context-specific strategies related to risk assessment, communication, problem solving techniques, preventive measures, and telehealth (*O'Sullivan et al., 2020*). Strategies that might be utilised to improve continuity of services were outlined in a report from the discussion by stakeholders involved in health system strengthening and Universal Health Coverage (UHC) in the WHO African Region. This discussion generated four main themes. Firstly, there is a necessity for collaborative planning, implementation, monitoring, and assessment of closely related priorities. For instance, a cholera outbreak should be addressed through collaboration with multiple sectors beyond health alone; by using water, sanitation, and hygiene (WASH) interventions. The second theme focused on the integration of health programmes, while the third theme stressed on the need to ensure that plans and ideas are moved from conceptualisation to implementation. This should be done in conjunction with prompt access to data on the most effective resilience strategies and information sharing. Lastly, it is critical to recognise the value of resilience and the need to build inherently resilient health systems along with the capacity to learn from shocks and improve future responses (*Tumusiime et al., 2020*).

### Health workforce

Two studies focused on strengthening the health workforce as part of health system resilience (*Witter et al., 2017*; *Tidwell & Razak, 2021*). In the first study, a health workforce programme was developed during the Ebola epidemic in Liberia, which increased the number and the expertise of their health workforce (doctors, nurses, and midwives) (*Tidwell & Razak, 2021*). The second study, which employed a mixed methods design, described the lived experiences of the health workforce across four countries names; Uganda, Sierra Leone, Zimbabwe, and Cambodia. This study sought to explore how healthcare providers coped with different shocks and crises, ranging from epidemics to economic and political crises. Authors found coping strategies such as measures taken to ensure one's physical safety (self-protection), psychosocial support from family and their community, and other practical strategies relating to work and remuneration (such as dual practice, supporting patients with their resources). Despite these, the authors stressed the need for more support and protection of the health workforce (*Witter et al., 2017*).

## DISCUSSION

There has never been a better opportunity for countries to embrace the idea of health system resilience than this time, when we are still recovering from the COVID-19 pandemic, one of the deadliest in the history of public health crises. The COVID-19 pandemic has further exposed the fragility of the health systems. This review, therefore, aimed to provide an overview of methods that countries, especially in Africa, might utilise to make their health care systems more resilient.

Most studies included in this review originated from North America, Europe, and Africa. Since Africa has been hit hard by Ebola, other infectious diseases, and humanitarian

crises for a long time, it is not surprising that the continent needs to strengthen the resilience of its health care system (*Gebremeskel et al., 2021*).

Strategies to boost health system resilience from our review cuts across all the health system blocks. Most studies from our review focused on areas related to service delivery, with authors stressing the need to keep vital services running even in times of crisis. The importance of ensuring the continuous provision of essential services cannot be overemphasised as countries are beginning to observe the impact of the disruption caused by the COVID-19 pandemic on essential services for tuberculosis, HIV/AIDS, and childhood immunisation (*World Health Organization, 2020*, *2021*; *Downey et al., 2022*).

In addition, the need for community health engagement was also highlighted by the authors. Community engagement promotes resilience, and community health professionals act as a bridge between communities and health systems. Furthermore, community buy-in or trust is crucial for resilient health systems (*Gebremeskel et al., 2021*). Community health workers play crucial roles in risk communication, contact tracing, transporting supplies, and ensuring the continuity of essential services (*Wilson Center, 2020*; *Gebremeskel et al., 2021*). CHWs are also an important link between the community and the response team and are crucial for building trust within the community. Furthermore, they are usually the primary source of information for the majority of people in remote areas in Africa and a crucial component of an effective response to emerging infectious diseases (*Wilson Center, 2020*). However, there are shortages of this cadre in Africa, requiring more training to increase their number, as shown in our study findings (*Tidwell & Razak, 2021*). Community health care workers are often underpaid, underutilised, and poorly integrated into health systems. This gap was exacerbated by the COVID-19 pandemic, which limited community engagement in health planning delivery, risk communication, and advocacy (*Wilson Center, 2020*). It is also critical that the primary healthcare system is not overlooked, as there are typically limited health services in rural areas than in urban areas (*Gebremeskel et al., 2021*).

The included articles made a strong case for strengthening the health workforce by increasing their numbers, providing all forms of support, including psychosocial support, and expanding their role in governance. A systematic review by *Ayanore et al. (2017)* reemphasizes the importance of policy shifts in African health systems that prioritise training a cadre of health care workers willing and able to provide timely responses to any disease or health threat.

Health financing is low in sub-Saharan Africa, and the poor health infrastructure in some African countries is attributable to a lack of long-term investment in health. The region lacks health infrastructure and has little funding for public health services (*Gebremeskel et al., 2021*). Our study noted that boosting strong financial infrastructure is crucial for health system resilience. A robust financial structure will play a significant role in achieving the already identified health system resilience strategies. Furthermore, investing in developing a robust and cost-effective surveillance capacity and establishing financial accountability in health financing and governance can help strengthen health system resilience (*Ayanore et al., 2017*). Such investments can potentially improve other areas, including access to medical products and diagnostics, protective equipment required

during health emergencies, the need for digitization of health information, and telemedicine and other innovative ideas, as highlighted in our study.

A study has shown the potential benefits of investing in robust information systems, including improved surveillance and disease tracking, timely detection and response to health system threats, improvement of treatment outcomes, and improving overall data quality (*Ayanore et al., 2017*).

The place of governance and leadership is also crucial. Our study highlights collaborations at local and international levels, creating policies beneficial to the health workforce and those that ensure that post-recovery plans are in place. A systematic review has further provided evidence on the need for health financing to deliver real-time health responses to health threats as a critical governance mechanism for boosting health system resilience in Africa. This review also supported our evidence on the need for creating partnerships to address any challenges countries face when facing health threats (*Ayanore et al., 2017*).

Health system resilience may be influenced by broader contextual factors. A multitude of contextual factors across various building blocks influence the resilience of health systems. For example, the health workforce's resilience may be significantly impacted by the working conditions and support structures established by effective governance, rather than solely relying on their skills and numbers. Good governance guarantees that health workers receive sufficient psychosocial support, equitable compensation, and opportunities for professional development, all of which are essential for their morale and ability to respond to crises. In the same way, the transparency and accountability that governance structures promote are essential for effective service delivery, as are community engagement and trust. Robust governance and leadership are also essential for the availability of financial resources, efficient health information systems, and access to essential medicines. Therefore, the governance and policy environments that underlie the health system's resilience can frequently be identified as the source of these variations across various building blocks.

## Policy implications of our study

This study provides several policy implications that can guide policymakers and health system managers in enhancing health system resilience. Firstly, there is a need for a substantial increase in health financing to ensure that health systems are adequately resourced to respond to emergencies. Policymakers should prioritize investments in health infrastructure, surveillance systems, and the health workforce. Secondly, the study highlights the importance of community health workers (CHWs) in bridging the gap between communities and health systems. Policies should focus on integrating CHWs into the formal health system, providing them with adequate training, fair compensation, and support. Thirdly, governance and leadership are crucial in ensuring the effectiveness of resilience strategies. Policymakers should foster collaborations at local and international levels, create policies that support the health workforce, and ensure that post-recovery plans are in place. Lastly, there should be an emphasis on continuous improvement and adaptability of health systems. Policymakers are advised to implement mechanisms for

regular review and adaptation of health policies to address emerging challenges and changing health landscapes. By addressing these areas, policymakers and health managers can use these findings to build more resilient health systems capable of withstanding future crises.

In terms of implications for research, more robust studies are needed to show resilience across all heat system blocks countries need to document their lessons for resilience. Additionally, studies such as systematic reviews which are more narrow-based are needed to assess the effectiveness of the resilience strategies.

Our study has some limitations. Firstly, we searched only two databases; thus, we may have missed some articles published in other databases including those not published in English language. In addition, the lack of consensus on the concept and definition of health system resilience (*World Health Organization, 2020*) may lead to the omission of eligible articles. We did not also synthesise evidence regarding the efficacy of these measures in promoting health system resilience as a systematic review would likely be the best approach for this. Furthermore, since our analyses were mostly based on deductive approach, this may present as a limitation as emergent themes that do not fit within the *a priori* framework may have omitted.

## CONCLUSION

The strategies that can improve health system resilience touch on all the health system blocks and do not ignore primary health care, which is often overlooked during crises. The latter requires context-specific strategies. Community engagement and the use of community healthcare workers to boost resilience are critical strategies requiring consideration. Our study shows that a resilient health system should be prepared for an impending crisis and have adaptable policies that will ensure adequate response at all levels and post-recovery plans. In addition, such health systems should be committed to continuous improvement. A substantial amount of health financing and governance structures are required to achieve these strategies. Further research is necessary to evaluate the effectiveness of these identified strategies in strengthening resilience in the health systems. Studying the effectiveness of these interventions would be a significant step towards implementation, especially in African settings where health systems are fragile.

## ACKNOWLEDGEMENTS

We wish to acknowledge the South African Medical Research Council for providing the equipment and the space to conduct the project.

### Funding

This study was funded by Cochrane South Africa (Grant number: 43500) and The World Health Organization African Region. The funders had no role in study design, data collection and analysis, decision to publish, or preparation of the manuscript.

## Grant Disclosures

The following grant information was disclosed by the authors:
Cochrane South Africa: 43500.
The World Health Organization African Region.

## Competing Interests

The authors declare that they have no competing interests.

## Author Contributions

- Dick Chamla conceived and designed the experiments, performed the experiments, authored or reviewed drafts of the article, and approved the final draft.
- Chinwe Iwu-Jaja conceived and designed the experiments, performed the experiments, analyzed the data, prepared figures and/or tables, authored or reviewed drafts of the article, and approved the final draft.
- Anelisa Jaca conceived and designed the experiments, performed the experiments, analyzed the data, authored or reviewed drafts of the article, and approved the final draft.
- Asiphe Mavi Ndlambe conceived and designed the experiments, performed the experiments, analyzed the data, authored or reviewed drafts of the article, and approved the final draft.
- Muyunda Buwa conceived and designed the experiments, performed the experiments, analyzed the data, authored or reviewed drafts of the article, and approved the final draft.
- Ngozi Idemili-Aronu conceived and designed the experiments, authored or reviewed drafts of the article, and approved the final draft.
- Joseph Okeibunor conceived and designed the experiments, authored or reviewed drafts of the article, and approved the final draft.
- Charles Shey Wiysonge conceived and designed the experiments, performed the experiments, analyzed the data, authored or reviewed drafts of the article, and approved the final draft.
- Abdou Salam Gueye conceived and designed the experiments, authored or reviewed drafts of the article, and approved the final draft.

## Data Availability

   This is a literature review.

## Supplemental Information

Supplemental information for this article can be found online at http://dx.doi.org/10.7717/peerj.17869#supplemental-information.

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
