# Peer review of "The critical elements of the health system that could make for resilience in the World Health Organization African Region: a scoping review"

_PeerJ, doi:10.7717/peerj.17869_

## Round 0.1 · original submission · Major Revisions

1. In addition to reviewers’ comments, it might be of value to rewrite the section on “rationale for conducting the study” for much clarity and consistency.
2. The discussions need additional work:
a. Could you discuss the broader contextual factors that contributed to variations in health system resilience across different building blocks? For instance, the resilience of health workers may hinge on the working conditions provided by effective governance.
b. What are the policy implications from this study and how do authors advise policymakers and managers to use these findings.

Reviewer 1 ·

Basic reporting

This study provides an insightful analysis of the crucial elements of the healthcare system in the African Region. It primarily addresses the impact of the COVID-19 pandemic and other infectious diseases that pose significant risks to public health. The findings of this scoping review can aid policymakers in devising efficient approaches to manage such situations. I must applaud the authors for their well-structured and well-executed study.

Experimental design

No comment

Validity of the findings

In comparison to the research conducted by Adamu AA et al. (Ref# 23) on the health system during the COVID-19 pandemic, what distinguishes your findings?

Reviewer 2 ·

Basic reporting

- Table 1 is very informative. But the discussions in section 3.3 summary of global evidence are too general. It’s hard to see which sentence is supported by which study, and what methods and results of the corresponding study. Could you elaborate on your arguments? Reference the studies for each of your sentences. Also describe how the studies were able to support that sentence.

Experimental design

- In "2.4 study selection section", please describe if the study selection was performed by two reviewers independently.
- It seems strange to include studies that were conducted outside of WHO African region. Could the findings of those studies be applied to the African region? Could you discuss the generalizability of your study findings? Could you elaborate on how the study findings from other regions could be translated to the African region in section 3.3?
- Could you also structure the section 3.4 into the 6 subsections like what you did for section 3.3? In addition, are there any extra aspects that you found from the 7 studies conducted in the African region that could be applied specifically to African regions? Could you add a subsection in section 3.4 to discuss that?

Validity of the findings

no comment

·

Basic reporting

• The review is well-written and easy to read and understand, even by a non-native English speaker like myself.

• The introduction sets the scene, highlights the importance of health system resilience, makes reference to relevant literature and provides a rationale for conducting this review.

Experimental design

• Since the article was sent for peer review, I suppose the editor checked it to ensure that it complied with the aims, scope, and instructions of the journal.

• The methods section needs substantial improvements to provide sufficient details and to ensure the transparency of the review process. Here are my comments:

- Comment #1: The authors chose the ‘scoping review’ method among various types of literature reviews. They should justify their choice of this method for reviewing literature on the subject under investigation.

- Comment #2: Arksey and O'Malley's seminal paper (2005) introduced a methodology for conducting a scoping study. Levac et al. (2010) and Daudt et al. (2013) later enhanced this framework. I recommend that the authors refer to this methodological guidance to improve the structure of the methods section in this review.

- Comment #3: The research question seems a bit vague. From a system-thinking perspective, all the components of a health system and their interactions are necessary to strengthen the system and, in this case, its resilience (de Savigny and Adam, 2009). The authors should instead focus on the interventions within each component (or building block) that contribute to the health system's resilience.

- Comment #4: The authors used the JBI's PCC framework but slightly modified it to exclude the population. To enhance transparency, I suggest that the authors provide a justification for this modification. Additionally, while the authors defined the Concept (health system resilience), the Context remains somewhat unclear, as it has not been explicitly defined. Hence, I would advise the authors to explicitly define what they mean by context for this review. From my understanding, the context here may be the COVID-19 pandemic and infectious disease outbreaks or overall health system crises.

- Comment #5: In the "Inclusion Criteria" sub-section, the author stated the JBI's PCC framework, defined the concept (health system resilience) and explained the frameworks used for data extraction and analysis (WHO’s building blocks and Palagyi et al.’s frameworks). However, the inclusion and exclusion criteria are not clearly specified. I suggest the authors list the eligibility (inclusion and exclusion) criteria based on specific factors such as paper content (concept and context), paper type, language, time period, etc. A table summarising these criteria would be helpful in aiding readers' comprehension.

- Comment #6: In line with my previous comment #4, I suggest that the "Types of sources" sub-section should be considered an eligibility (inclusion/exclusion) criteria parameter rather than a step of a scoping review. Additionally, the sentence "Studies published in the English language were included with no date limitations" can be relocated to the eligibility (inclusion/exclusion) criteria sub-section for clarity.

- Comment #7: The PRISMA-ScR flowchart lacks clarity regarding the reasons for exclusion, particularly at the eligibility step, as the exclusion criteria are not explicitly mentioned in the text. It would be beneficial if the authors included details of the reasons for exclusion in the flowchart. Additionally, I suggest that the authors provide the PRISMA-ScR checklist as a supplemental file to ensure compliance with the required standards for reporting scoping review.

- Comment #8: The author extracted and analysed data using the WHO’s building block framework. For more transparency, I recommend that the authors justify their choice of this framework among other health system resilience frameworks.

- Comment #9: The sub-section on "data analysis" is not very detailed. The authors seem to have adopted a deductive approach to summarising their findings by using the WHO’s building blocks framework a priori. However, this deductive analysis may overlook emergent themes that don't fit within the a priori framework. To ensure transparency, I recommend that the authors mention how they addressed such situations or acknowledge them as a limitation of the study.

- Comment #10: There are two overlapping in-text referencing systems: one with brackets ( ) and the other with square brackets [ ] (see lines #122-133). Please check and correct this.

Validity of the findings

The findings of this review also need improvement in presentation and depth. Below are my comments related to the results and discussion sections of the manuscript.

- Comment #11: I recommend that the authors summarise their findings in tables, as it will help readers to understand the information more easily. For instance, they can summarise the characteristics of the included papers in a table that lists their frequency (numbers and percentages) and references. They can also summarise the distribution of papers by health system building block and health system resilience strategies. This will help readers quickly understand which building blocks or resilience strategies are currently being more focused on in current practices.

- Comment #12: It would be beneficial if the authors provided evidence (if any) of how interventions affect health system resilience, rather than just listing them under each building block.

- Comment #13: It's important to note that even though the scoping review doesn't require appraising the quality of included papers, this lack of appraisal should be considered when interpreting the results. Additionally, the fact that the review only included papers published in English is a limitation, as the author may have missed relevant literature published in other languages.

- Comment #14: The authors mentioned that they did not examine the efficacy of the measures that promote health system resilience. This statement appears to contradict the review's title, which examines the critical components of the health system that can contribute to resilience in the African Region. As per my comments #3 and #12, I suggest that the authors analyse the efficacy of the interventions or measures identified in this review in promoting health system resilience. This will make the review more robust and impactful.

Additional comments

No comment

---

## Round 0.2 · accepted · Accept

Authors have comprehensively addressed peer review comments and this paper now merits a publication.